# Safety Evaluation, Biogenic Amine Formation, and Enzymatic Activity Profiles of Autochthonous Enterocin-Producing Greek Cheese Isolates of the *Enterococcus faecium/durans* Group

**DOI:** 10.3390/microorganisms9040777

**Published:** 2021-04-08

**Authors:** Charikleia Tsanasidou, Stamatia Asimakoula, Nikoletta Sameli, Christos Fanitsios, Elpiniki Vandera, Loulouda Bosnea, Anna-Irini Koukkou, John Samelis

**Affiliations:** 1Dairy Research Department, General Directorate of Agricultural Research, Hellenic Agricultural Organization ‘DIMITRA’, Katsikas, 45221 Ioannina, Greece; xaroulatsan@gmail.com (C.T.); nikol.sameli@gmail.com (N.S.); louloudaBosnea@gmail.com (L.B.); 2Laboratory of Biochemistry, Department of Chemistry, University of Ioannina, 45110 Ioannina, Greece; s.asimakoula@uoi.gr (S.A.); chris.fanitsios@gmail.com (C.F.); evandera@uoi.gr (E.V.); akukku@uoi.gr (A.-I.K.)

**Keywords:** *Enterococcus faecium*, *E. durans*, enterocins A, B, P, virulence genes, antibiotic resistance, biogenic amines, cheese adjuncts, Graviera, Galotyri

## Abstract

Autochthonous single (Ent+) or multiple (m-Ent+) enterocin-producing strains of dairy enterococci show promise for use as bioprotective adjunct cultures in traditional cheese technologies, provided they possess no pathogenic traits. This study evaluated safety, decarboxylase activity, and enzymatic (API ZYM) activity profiles of nine Ent+ or m-Ent+ Greek cheese isolates previously assigned to four distinct *E. faecium (*represented by the isolates KE64 *(entA),* GL31 *(entA)*, KE82 *(ent*A*-ent*B*-ent*P) and KE77 *(ent*A*-ent*B*-ent*P*-bac31)*) and two *E. durans* (represented by the isolates KE100 *(ent*P*)* and KE108 *(ent*P*-bac31-cyl*)) strain genotypes. No strain was *β*-hemolytic or harbored *van*A and *van*B or the virulence genes *agg, ace, espA, IS16, hyl*, or *gel*E. All strains were of moderate to high sensitivity to ampicillin, ciprofloxacin, chloramphenicol, erythromycin, gentamicin, penicillin, tetracycline, and vancomycin, except for the *E. faecium* KE64 and KE82 strains, which were resistant to erythromycin and penicillin. All cheese strains showed moderate to strong esterase-lipase and aminopeptidase activities and formed tyramine, but none formed histamine in vitro. In conclusion, all Ent+ or m-Ent+ strain genotypes of the *E. faecium/durans* group, except for the *cyl*-positive *E. durans* KE108, were safe for use as adjunct cultures in traditional Greek cheeses. Further in situ biotechnological evaluations of the strains in real cheese-making trials are required.

## 1. Introduction

Autochthonous dairy enterococci are strongly associated with traditional cheeses of the Mediterranean area [1,2,3]. Their levels range from 10^4^ to 10^6^ CFU/g in the fresh cheese curds and from 10^5^ to 10^8^ CFU/g in the resultant ripened cheeses, depending on the cheese type, whether the milk is processed raw, thermized, or pasteurized and on the type and activity of starter lactic acid bacteria (LAB) culture added to the milk [1,3,4]. Particularly in commercial starter culture (CSC)-free, traditional Greek hard and semi-hard cheeses (pH > 5.0–5.7) produced from raw, thermized, or open-batch pasteurized milk, members of the *E. faecium/durans* and *E. faecalis* groups, are naturally selected to prevail [1,3,4]. Their prevalent growth in the fresh curds, and quite often during ripening of artisan Greek hard cheeses, has been attributed to the fact that enterococci are more tolerant to milk thermization and pasteurization than other raw milk LAB biota [5] and also have a broader growth temperature range (4–45 °C), with an optimum at 37 °C [6]. Growth of enterococci is more restricted in traditional Greek fresh or ripened acid-curd cheeses (pH 3.8–4.7) [3,7] and soft white cheeses ripened in brine, such as Protected Designation of Origin (PDO) Feta cheese (ca. pH 4.5) [3], because they are sensitive to (lactic) acid. In general, beneficial dairy enterococci contribute to the aroma and taste of mature traditional cheeses, mainly by their in situ proteolytic activities [2,8,9]. In addition, numerous *Enterococcus* strains of (dairy) food origin boast production of a heterogeneous group of ribosomally synthesized antimicrobial peptides or proteins, termed enterocins, contributing to inhibit growth of several pathogenic and spoilage bacteria in situ in cheese [1,2,10]. Moreover, several single (Ent+) or multiple (m-Ent+) enterocin-producing dairy *Enterococcus* strains possess various probiotic properties [11]. Thus, these strains may offer health benefits and are required for developing novel functional (dairy) foods, such as probiotic cheeses [8,11,12].

On the other hand, there are still reasonable concerns and continuing scientific controversy regarding the safety risks associated with the natural prevalence and commercial application of enterococci in fresh or fermented (dairy) foods [1,8,12,13]. Overall, the genus *Enterococcus* includes opportunistic human pathogenic species or strains, many of which are β-hemolytic and/or multi-drug resistant clinical bacteria that possess and express virulence genes in vivo and can also transfer antibiotic resistance genes to LAB and other beneficial food bacteria in situ [12,13,14]. Moreover, enterococci are able to grow at alkaline pH and produce high levels of biogenic amines by decarboxylating the amino acids present in (dairy) foods [8,15]. Hence, the use of *Enterococcus* spp. as commercial adjunct cultures in the food (dairy) industry is still prohibited because the genus has neither a generally regarded as safe (GRAS) status nor has it been included in the qualified presumption of safety (QPS) list [8]. Only certain *E. faecium* and *E. faecalis* strains have so far been permitted for use in dairy foods as individual co-starter cultures or have an approval as probiotic food supplements [2,8,11,12]. Nonetheless, an increasing number of recent polyphasic cheese ecology studies, including biotechnological and safety evaluations of numerous autochthonous harmless *Enterococcus* spp. isolates, have weakened safety concerns associated with the natural prevalence of dairy enterococci in artisan Mediterranean and Western Balkan cheese products [16,17,18,19,20].

Recently, we confirmed that CSC-free, naturally ripened traditional Greek Graviera cheeses represented a specific food ecological niche dominated (79%) by diverse strain genotypes of the *E. faecium/durans* genomic group; 17% of those isolates exhibited Ent+ antilisterial activity in vitro [4]. The isolation frequency of *E. faecalis* from those Graviera cheeses was 3% only [4]. Conversely, *E. faecalis* was more prevalent in artisan Galotyri PDO acid-curd cheeses from the Epirus market [7]. Eventually, 15 selected Ent+ *Enterococcus* isolates from Graviera and Galotyri cheeses were genotyped, and their structural enterocin gene profiles and mode of antilisterial activity were determined [6]. Most Ent+ *E. faecalis* isolates from Galotyri cheese harbored the cytolysin (*cylL_L_*) gene [21], while one *E. faecalis* strain genotype was strongly β-hemolytic [6]. Thus, regardless of their strong antilisterial activity, *E. faecalis* isolates from Galotyri cheese were a priori excluded from (dairy) food applications [6]. Conversely, none of the Ent+ *E. faecium* and *E. durans* Greek cheese isolates were β-hemolytic. Five of them possessed only *ent*A or *ent*P genes, while three *E. faecium* and one *E. durans* isolates were m-Ent+ gene possessors [6]. The present is a follow-up of the Vandera et al. [6] study, aiming to evaluate whether the above promising antilisterial strains of the *E. faecium/durans* genomic group are completely harmless and technologically suitable to be applied as co-starter or protective adjunct cultures in traditional Greek cheese technologies.

## 2. Materials and Methods

### 2.1. LAB Strains and Culture Conditions

Nine autochthonous Ent+ or m-Ent+ isolates of the *E. faecium/durans* group from artisan Greek Graviera and Galotyri PDO cheeses were selected for this study. Their designation, species identity, structural enterocin gene profiles, and mode of antilisterial activity data, adapted from previous studies, are summarized in Table 1. Briefly, the nine antilisterial cheese isolates belonged to four (EF1–EF4) and two (ED1,ED2) distinct strain genotypes of *E. faecium* and *E. durans*, respectively (Table 1). Their Random Amplified Polymorphic DNA (RAPD)-based genotyping was consistent with their biotyping at the strain level, except for *E. faecium* KE82 and KE118 isolates, which were biochemically distinct strains despite the fact that they shared their m-Ent+ (*ent*A*-ent*B*-ent*P) and RAPD plasmid profiles (Table 1).

Six additional LAB strains were used as controls in the present bioassays (Table 1). In specific, *E. faecalis* ATCC^®^ 29212™, a reference strain that possesses the *gel*E and *ace* virulence genes [22,23], and *E. faecium* 315VR, a *van*A-positive isolate from a patient at the University Hospital of Ioannina, kindly provided by the Microbiology Laboratory of Medical Department, University of Ioannina, served as controls for the virulence gene detection and antibiotic susceptibility assays. Additional details on the origin, isolation, and primary properties of the Ent-negative *E. faecium* KE85, *S. thermophilus* CSL-ST1, *Lc. lactis* subsp. *cremoris* M78, and *Lactiplantibacillus* (formerly *Lactobacillus*) *plantarum* H25 starter or adjunct control strains are provided in the respective references in Table 1.

All LAB strains were incubated at 30 °C for 24 h in MRS broth (Neogen Culture Media, LAB094, Heywood, UK) for two sequent transfers before use, except for *S. thermophilus* ST1, which was cultivated twice in M17 broth (Merck, Darmstadt, Germany) at 37 °C [24].

### 2.2. Hemolytic Activity

All LAB isolates were tested comparatively for hemolytic activity on ready-to-use 5% sheep blood agar plates (Bioprepare, Attiki, Greece), incubated aerobically at 37 °C for 48 to 72 h [6,19,20]. One single batch of blood agar plates was used under identical experimental conditions to ensure uniformity of the results. The control starter strains *S. thermophilus* ST1, *Lc. lactis* subsp. *cremoris* M78, and *Lb. plantarum* H25 were expected to display a γ-hemolytic (negative) reaction. The virulent *E. faecalis* ATCC 29212 and the clinical *E. faecium* 315VR strains were evaluated for their potent β-hemolytic activity.

### 2.3. Antibiotic Susceptibility

Antibiotic susceptibility was tested by the Kirby–Bauer disk diffusion method [25]. Ready-to-use antibiotic disks were purchased from BioMaxima SA (Lublin, Poland). All LAB strains were screened for resistance to eight common antibiotics amongst those specified by the Clinical and Laboratory Standards Institute (CLSI) [26] for enterococci (μg per disk): ampicillin (AMP; 10), chloramphenicol (CHL; 30), ciprofloxacin (CIP; 5), erythromycin (ERY; 15), gentamicin (GEN; 10), penicillin G (PEN; 10 units per disk), tetracycline (TET; 30), and vancomycin (VAN; 30) as follows: 100μL of fresh (24-h) M17 broth cultures of each strain were spread with a sterile glass rod on two individual M17 agar plates, which were left to absorb for 5 to 10 min. Afterward, 3–4 disks with different antibiotics were placed over the LAB cell lawn in each plate. The net diameter of the inhibition zones formed around each antibiotic disk was measured with a micrometer (model D15, Mitutoyo, Kanagawa, Japan) after overnight incubation of the M17 agar plates at 37 °C for all strains. To account for potential LAB media effects on the size and clearness of the inhibition zones, the antibiotic susceptibility test was repeated, as above, by seeding the fresh cell culture lawns on MRS and BHI agar plates, comparatively to the M17 agar plates. The results were interpreted according to the breakpoints recommended by CLSI [26] for enterococci. *E. faecium* 315VR and *E. faecalis* ATCC 29212 (vancomycin-sensitive) were used as positive and negative control strains, respectively.

### 2.4. Detection of Vancomycin Resistance and Virulence Genes by PCR

Bacterial genomic DNA (gDNA) from overnight cell cultures in MRS broth was extracted using cetyltrimethylammonium bromide (CTAB), according to William et al. [28] v. 3 and served as a template in PCR amplification reactions for the safety evaluation of the strains. All PCR reactions were performed in a DNA thermal cycler (G-STORM GS1, Labtech International Ltd., UK), using Taq Polymerase (KAPA Taq PCR Kit, Kapabiosystems Inc., Wilmington, MA, USA), according to the manufacturer’s instructions. Agarose gel electrophoreses were carried out using standard methodology [29]. The primers used were synthesized by Eurofins Genomics Germany GmbH (Ebersberg, Germany).

The strains were assessed for the presence of vancomycin resistance genes (*van*A*, van*B) and six virulence genes (*agg*, *ace*, *esp*A, IS*16*, *hyl*, and *gel*E) using specific primers (Table 2). Selection of the above virulence genes was based on the European Food Safety Authority (EFSA) guidance on the safety assessment of *E. faecium* in animal nutrition [30]. The presence of structural genes responsible for vancomycin resistance was investigated with a multiplex PCR reaction method, as described by Petrich et al. [31]. *E. faecium* 315VR was used as a positive control. The presence of virulence genes was investigated by PCR reactions for each target gene, as previously described [31,32,33,34,35]. *E. faecalis* ATCC 29212 served as a positive control strain for the virulence genes tested.

### 2.5. Detection of Biogenic Amine Formation

The Ent+ or m-Ent+ *Enterococcus* isolates were evaluated for their in vitro ability to produce the commonest biogenic amines (BAs) in cheese, histamine (HIS), and tyramine (TYR) by using the screening method of Bover-Cid and Holzapfel [36] without the previous activation step. Briefly, 100 μL of fresh (24-h) MRS or M17 pure cultures of the strains were inoculated in 5-mL portions of improved BA broth medium with 1% histidine or 1% tyrosine (Merck), adjusted to pH 5.4 prior to autoclaving. Tubes with 5-mL portions of control (CN) BA broth, without histidine or tyrosine, were inoculated for comparison. All cultures were incubated at 37 °C for 72 h. Positive tubes showing BA (either HIS or TYR) production were recorded by the color change of the bromocresol indicator from yellowish to bright blue or purple. *E. faecalis* ATCC 29212, *E. faecium* 315VR, and the non-enterococcal LAB strains ST1, M78, and H25 (Table 1) were also tested as above. The in vitro formation of TYR and HIS in the pure BA broth cultures of most LAB strains was verified and quantified by HPLC after completion of the 72-h incubation period.

### 2.6. Determination of Biogenic Amine Formation by HPLC

Determination by HPLC of the BAs in the LAB culture broth supernatants was carried out by acid extraction and derivatization using the method of Eerola et al. [37]. To account for the potential formation of HIS or TYR in the CN samples and evaluate the ability of the strains to form additional BAs, six BAs in total, namely cadaverine, histamine, putrescine, spermidine, tryptamine, tyramine, and 2-phenylethylamine, purchased from Sigma-Aldrich Chemie GmbH (Steinheim, Germany) or Acros Organics (Geel, Belgium), were analyzed by HPLC.

For sample preparation, the cell-free cultured (37 °C; 72 h) filtrate supernatants of each LAB strain that originally contained 1% histidine or 1% tyrosine, respectively, were pooled, regardless of whether they were positive or negative. All respective CN culture supernatants stayed separate for HPLC analysis. Afterward, 4 mL of each LAB culture, together with 125 μL of each BA stock solution, were adjusted to 25 mL with HClO_4_ 0.4 M. Following derivatization according to [37], the final sample mixture was diluted to 5 mL with acetonitrile (Carlo Ebra Reagents S.A.S, Val de Reuil, France), centrifuged at 2500 rpm at 25 °C for 5 min, and 20 μL of the supernatant was injected into the HPLC.

The BA derivatives (20 μL) filtered through a 0.45-μm filter were analyzed on a LC-20AT high-performance liquid chromatograph (Shimadzu, Tokyo, Japan) equipped with a thermo-stated auto-sampler (SIL-20A), a high pressure mixing binary pump (LC-20AT), a column oven (CTO-20A), and a diode array detector (SPD-M20A). Separation of the derivatives was carried out on a Shim-pack GIST C18 column (3 μm, 100 × 3 mm I.D, Shimazdu, Japan) equipped with a guard column. A gradient elution program with ammonium acetate (Carlo Erba) 0.1 M (A) and acetonitrile (B) was used. Gradient started at 50% and ended at 90% acetonitrile in 19 min. The flow rate of the mobile phase was 0.9 mL/min, the column temperature was set at 40 °C, and the peaks were detected at 254 nm. Identification of the BAs was based on their retention times. The lines of regression calculated were used to compute the amount of the analytes in the samples by interpolation, using as an internal standard, 1 mg/mL of 1,7-diaminoheptane (Sigma-Alrdich). The HPLC system was equilibrated for 10 min before the next analysis.

### 2.7. Enzymatic Activity Profiles

The enzymatic activity profiles of the Ent+ or m-Ent+ *Enterococcus* and the control LAB strains (Table 1) were determined by the API ZYM method (BioMerieux, Marcy l’ Etoile, Lyon, France) at 37 °C for 4 h, according to the manufacturer’s instructions.

## 3. Results

### 3.1. Safety Characteristics of the Ent+ or m-Ent+ E. Faecium and E. Durans Cheese Isolates

None of the cheese LAB isolates tested in this study were β-hemolytic. However, all nine Ent+ or m-Ent+ *E. faecium* or *E. durans* isolates and the Ent-negative KE85 strain showed α-hemolysis manifested as a green discoloration around the streaked colony growth on 5% sheep blood agar (Table 3). *E. durans* KE96 = KE100 showed the weakest α-hemolytic reaction that might have been perceived as γ-hemolysis in the absence of the *S. thermophilus* ST1 and *Lc. lactis* M78 control strains, which clearly were γ-hemolytic (no color change). In contrast, *Lb. plantarum* H25 displayed the strongest α-hemolytic reaction. The two virulent *Enterococcus* control strains were β-hemolytic (Table 3).

All Ent+ or m-Ent+ *E. faecium* and *E. durans* cheese isolates were susceptible to ampicillin, chloramphenicol, tetracycline, and, most importantly, vancomycin. *E. faecium* KE85 was also susceptible to ampicillin, chloramphenicol, and vancomycin, but it was resistant to tetracycline (Table 3). Susceptibility of the isolates to the above four antibiotics was measured according to the disk-assay breakpoints recommended by CLSI [26] for enterococci and was in comparison to the corresponding susceptibilities of the *van*A*+ E. faecium* 315VR positive control. Actually, this clinical strain, apart from being β-hemolytic, was found to be extremely (no inhibition zone) to highly multi-resistant (<10 mm) to seven antibiotics tested; it was susceptible to chloramphenicol only (Table 3). *E. faecium* KE64 = KE67, KE82, and KE85 strains were resistant to penicillin, whereas the remaining three *E. faecium* KE77, KE118, and GL31 strains and both strain genotypes of *E. durans* were susceptible to penicillin. Additionally, according to the CLSI protocol, all *Enterococcus* cheese isolates were resistant or of intermediate susceptibility to ciprofloxacin and erythromycin, despite being inhibited by the above antibiotics (11.0–20.9 mm), and none were highly resistant (no inhibition zone), while the 315VR clinical strain was (Table 3).

Notably, the Ent+ or m-Ent+ *E. durans* strains were also more susceptible to ciprofloxacin, and mainly eryrthomycin, than the m-Ent+ *E. faecium* KE82 and KE118 and the EntA+ *E. faecium* KE64 = KE67 strains (Table 3). Compared to *E. faecium* 315VR, all *E. faecium* and *E. durans* cheese isolates were susceptible to only 10 μg/disk of gentamicin. Finally, *S. thermophilus* ST1 was susceptible to all antibiotics tested. *Lc. lactis* M78 was phenotypically resistant to ciprofloxacin, gentamicin, and tetracycline, and *Lb. plantarum* H25 was phenotypically resistant to ciprofloxacin, gentamicin, and vancomycin (Table 3).

None of the cheese starter or adjunct LAB strains evaluated in this study, including all nine Ent+ or m-Ent+ *E. faecium* and *E. durans* isolates, possessed any of the *van*A, *van*B, *agg, ace, esp*A, IS*16*, *hyl* or *gel*E genes (Appendix A).

All *E. faecium* and *E. durans* cheese isolates were profoundly able to form tyramine in vitro; however, none formed histamine under the same culturing conditions (Table 4). Tyramine formation was confirmed by HPLC quantifications, which showed that the m-Ent+ (A-B-P) *E. faecium* KE82 was the highest tyramine-producer, whereas the EntA+ *E. faecium* GL31 was the lowest tyramine-producer. Overall, the *E. durans* strains formed less amounts of tyramine than the *E. faecium* strains in BA broth in vitro (Table 4).

None of the ST1, M78, or H25 non-enterococcal, starter, or adjunct LAB strains formed tyramine or histamine in vitro. Additionally, minor tyramine and histamine amounts were formed in four of the respective cultured BA control samples (KE77, KE108, M78, and H25) tested by HPLC (Table 4). Furthermore, minor to negligible amounts of additional BAs, mainly tryptamine and 2-phenylethylamine were detected in all HPLC-tested cultured BA samples of the LAB strains, irrespective of the presence of 1% tyrosine and 1% histidine as precursor amino acids (data not shown).

### 3.2. Species-Dependent Differences of the Cheese Isolates in Their Enzymatic Activity Profiles

The API-ZYM activity profiles of the Ent+ or m-Ent+ cheese isolates are shown in Table 5, comparatively with the respective profiles of the control ST1, M78, H25, and KE85 strains. Major similarities, but also major species-dependent or strain-dependent differences, were observed. All *Enterococcus* isolates, including the Ent-negative KE85 control strain showed generally moderate alkaline phosphatase activity and moderate to high esterace (C4) and esterase-lipase (C8) activities, whereas no isolate showed lipase (C14) activity. Additionally, all *Enterococcus* isolates displayed acid phosphatase and napthol-AS-BI-phosphohydrolase activities, except the Ent- KE85 control strain (Table 5). With regard to their proteolytic activities, all amino acid (i.e., leucine, valine, and cystine) arylamidase reactions were moderate to strong in all Ent+ or m-Ent+ cheese isolates. Again, the only exception was the Ent- KE85 control strain, which was positive with leucine but negative with valine and cystine (Table 5).

Prominent differences occurred in α-chymotrypsin activity; they appeared to be species-dependent and secondarily strain-dependent. Indeed, both *E. durans* strain genotypes lacked α-chymotrypsin activity. The m-Ent+ *E. faecium* KE118 strain lacked α-chymotrypsin activity also. In contrast, the other m-Ent+ *E. faecium* strain KE82 that shared its structural *ent*A*-ent*B*-ent*P gene and RAPD profiles with strain KE118 (Table 1) was strongly active for α-chymotrypsin. This result confirmed that KE82 and KE118 are clearly different strain biotypes. The remaining *E. faecium* strains displayed weak α-chymotrypsin activity, including the Ent-KE85 control strain, which, overall, possessed the poorest enzymatic activity profile (Table 5).

All enterococcal strains displayed negative glycolytic enzyme reactions, except the m-Ent+ *E. faecium* KE77 strain, which showed weak α-galactosidase and α-glucosidase activities and the EntA+ *E. faecium* KE64 = KE67 strain genotype, which showed weak β-galactosidase activity (Table 5). The *E. durans* Graviera cheese isolates (KE96, KE100, and KE108) were negative for α-galactosidase (Table 5). In summary, the API-ZYM profiles provided several enzymatic reactions for the differentiation of *E. faecium* and *E. durans* cheese isolates at both the species and strain level. Contrary to the *E. faecium/durans* group of cheese isolates, the starter and adjunct control (non-enterococcal) strains displayed strong α-galactosidase (H25), β-galactosidase (ST1, H25), and α-glucosidase (M78, H25) activities (Table 5). Clearly, the nonstarter *Lb. plantarum* H25 possessed the most and overall strongest enzymatic activities, followed by the indigenous *nis*A*+ Lc. lactis* subsp*. cremoris* M78 co-starter. Conversely, the primary starter *S. thermophilus* ST1 showed only strong leucine arylamidase, acid phosphatase, and β-galactosidase and moderate esterase (C4) activities, while it was the only strain that lacked alkaline phosphatase activity. Finally, none of the LAB strains tested during this study showed trypsin or α-mannosidase or α-fucosidase activities (Table 5).

## 4. Discussion

The lack of β-hemolytic activity and the absence of six common virulence genes and the two vancomycin resistance genes, *van*A and *van*B (Table 2), from the genome of all nine autochthonous, Ent+ or m-Ent+, cheese isolates of *E. faecium* and *E. durans*, were the most important and encouraging results of this study regarding their safe use as adjunct cultures in traditional Greek dairy products. Particularly for *E. faecium*, the absence of three virulence markers, the IS*16*, *esp*, and *hyl*-like genes, associated with clinical strains of the species should be a prerequisite for strains to be considered safe for use as feed additives in animal nutrition by the EFSA [30]. This requirement is logically extended to *E. faecium* strains intended for use in foods. Additionally, vancomycin resistance, depicted by the single or combined possession of several *van* genes, is one of the most common virulent traits in clinical enterococci, particularly in *E. faecium* strain genotypes associated with nosocomial infections [11,12]. Unfortunately, since 2000, the isolation of vancomycin-resistant *E. faecium* strains from food processing environments and products, including traditional cheeses, has increased globally [1,8,13]. Particularly in the Mediterranean area, one of the first alarming screening studies was conducted by Giraffa et al. [38]: 25% of 102 *Enterococcus* isolates from industrial and artisan Italian cheeses were found to be resistant to both vancomycin and teicoplanin. Most of them belonged to *E. feacium,* harbored *van*A, and showed a high level of vancomycin resistance (MIC 128–512 μg/mL). However, fortunately, a considerable number of later studies showed that the incidence of virulence factors in food (cheese) isolates of *E. faecium* is strain specific, and, overall, much fewer strains within an isolated food group, most of which belonged to *E. faecalis*, harbored the IS*16*, *esp*A, *hyl*-like, and/or the *agg, ace*, and *gel*E genes [16,20,39,40,41,42,43,44]. In specific, Ben Omar et al. [42] reported no detection of virulence traits (hemolysin, gelatinase, or DNAse activities) and no possession of virulence genes, including *cyl*L, *ace*, *asa*I, *esp*, and *van*A/*van*B, amongst 50 selected *Enterococcus* strains from Spanish dairy, meat, and vegetable foods, while only one *E. faecium* strain was resistant to vancomycin and teicoplanin. Limited occurrence of virulence, vancomycin, and other antibiotic resistance genes were also found in *E. faecium* isolates from equipment surfaces, raw materials, and traditional cheeses in Italy and Portugal [41], home-made white brine Bulgarian cheeses [43], and during traditional Portuguese Terrincho cheese making [44]. Gaglio et al. [41] specified three strains from *E. faecium*, *E. durans*, and *E. casseliflavus*, respectively, isolated from stretched Italian cheeses that were totally free of virulence determinants, as were all Ent+ or m-Ent+ *E. faecium* and *E. durans* Greek cheese isolates of this study.

Moreover, in general agreement with our previous findings regarding the natural presence of β-hemolytic and *cyl*-positive *E. faecalis* strains in Galotyri PDO cheeses [6], the cytolysin gene/s and some of the virulence genes in Table 2 were detected in several *E. faecalis* cheese isolates [16,19,20]. However, these genes were absent in *E. faecium* isolates from Italian Valtelina Casera raw milk cheese [16] or they were sporadically detected in *E. faecium* and *E. durans* isolates from Istrian raw milk cheeses [20], other Serbian or Croatian cheeses [17,45], and Egyptian fresh raw milk cheeses [46]. Similarly, few *Enterococcus* isolates, most likely *E. faecalis* rather than *E. faecium*, from raw donkey milk in Cyprus, harbored the *gel*E, *asa*1, *ace*, and *esp* genes, but none of the *van*A or *van*B genes [47]. No *Enterococcus* isolate from Pico cheese presented *van*A or *van*B genes, while 93% were α-hemolytic [19], in general agreement with our present findings (Table 3).

To summarize the *Enterococcus* safety part of this study, selection for testing of the IS*16*, *esp*A, *hyl, agg, ace,* and *gel*E virulence genes plus the *van*A and *van*B genes (Table 2) was based on the EFSA guidance [30] but also on a preceding literature survey, including all previous dairy food studies by other workers discussed above. Absence of all eight genes tested (Table 2) and of the cytolysin gene from any *Enterococcus* sp., including *E. faecium* or *E. durans*, candidate strain for use as adjunct culture in (dairy) foods is required. However, this requirement appears not to apply regarding the presence of the *esp* gene (previously correlated with pathogenesis) in food (dairy) *Enterococcus* strains: this gene mostly associates with the genes involved in adhesion properties and biofilm formation, implicating their beneficial probiotic role in gut colonization as probiotics, rather than with the virulence traits [45,47]. Accordingly, *E. faecium* 894 from artisanal goatskin casing Turkish Tulum cheese was shown to be a probiotic candidate strain for further in vivo studies, although it harbored *asa*1, *gel*E, and *cyl*A genes [48]. Notably, the cytolysin gene was also detected in our m-Ent+ *E. durans* KE108 strain genotype (Table 1).

Nevertheless, compared to *E. faecium*, *E. durans* is less studied regarding its biotechnological and safe use in cheese and other dairy products. However, *E. durans* was the most prevalent beneficial species in several commercial batches of naturally fermented and ripened Greek Graviera cheeses [4] and in many Western Balkan artisan cheeses studied more recently [17]. Thus, 10 completely harmless and biotechnologically promising *E. durans* strains from the above Western Balkan cheeses were selected for use as cheese adjunct cultures [17]. *E. durans* LAB18s, another safe strain from Brazilian Minas Frescal cheese, without virulence or vancomycin resistance genes and susceptible to erythromycin, tetracycline, vancomycin, gentamicin, and penicillin, has been described [49].

The in vitro phenotypic susceptibility of cheese isolates of the *E. faecium/durans* genomic group to different antibiotic groups is also a strongly strain-specific trait that displays major variations from country to country, from the raw materials to the final cheese product, and between cheese varieties and research studies. Overall, apart from their susceptibility to vancomycin discussed above, most dairy isolates of *E. faecium* are also generally susceptible to ampicillin, chloramphenicol, gentamicin, penicillin, and tetracycline [39,43,44,47,48]. However, many *E. faecium* strains that display resistance to gentamicin, penicillin, and mainly tetracycline and present the respective antibiotic resistance genes occur in traditional cheeses [20,41,46]. Conversely, most *E. faecium* isolates from (dairy) foods display an increased resistance or intermediate susceptibility to erythromycin and ciprofloxacin [16,17,20,39,41,47]. In this study, none of our antagonistic *E. faecium* isolates was extremely resistant to the above antibiotics, as the clinical 315VR strain was. However, most of them showed a moderate resistance or intermediate susceptibility to erythromycin and ciprofloxacin, while three were resistant to penicillin and *E. faecium* KE85 was distinctly resistant to tetracycline (Table 3). *E. faecium* isolates with multi-resistance profiles, i.e., from 2 to 4 of the above antibiotics, were also frequent in cheese studies by others [17], while *E. durans* isolates presented an overall higher antibiotic susceptibility than their *E. faecium* co-isolates [17,49], as they also did in Table 3.

The strong in vitro tyrosine decarboxylase activity and the very weak, if any, histidine decarboxylase activity of all Greek cheese isolates of the *E. faecium/durans* group (Table 4) are in complete agreement with the data of many relevant previous studies. Bover-Cid and Holzapfel [36] screened 10 *E. faecium,* 1 *E. durans,* and 15 *E. faecalis* for BA formation in improved broth, followed by HPLC quantification, and found tyramine levels from 379 to 4986 mg/L, depending on the strain rather than the species, whereas no strain formed histamine. Later, Ladero et al. [50] suggested that tyramine biosynthesis is actually a species-level characteristic trait of *E. faecium*, *E. durans*, and *E. faecalis*, which we also confirmed in this study. Similarly, levels of tyramine ranged from 99.1 to 3664.7 mg/L within eight strains of the *E. faecium/durans* group cultured in modified BA broth media by Espinosa-Pesqueira et al. [51]. Notably, both tyramine and histamine were not detected in the cultures of four strains of *Lb. plantarum*, one *Lc. Lactis*, and one *S. thermophilus* grown in MP decarboxylase broth by Coton et al. [52], which corroborates our findings regarding the control strains H25, M78, and ST1, respectively (Table 4). Nevertheless, biogenic amines, particularly tyramine, histamine, putrescine, and cadaverine, are naturally formed by decarboxylating bacteria, including enterococci, and yeasts in all dairy products. However, their levels readily increase >200 mg/Kg in mold-ripened and hard/semi-hard ripened raw milk cheeses, where they may occasionally increase >1000 mg/Kg to cause toxicological concerns [15]. On the contrary, BAs remain undetected or below 60 mg/Kg in milk, fresh cheese curds, whey, and unripened cheeses from pasteurized milk [15]. Pilot-scale Galotyri and Graviera cheese trials are in progress to evaluate the ability of selected Ent+ *E. faecium/durans* adjuncts to form tyramine in situ.

Finally, the hydrolysis (API-ZYM) profiles from whole cells of the nine Ent+ or m-Ent+ *E. faecium* and *E. durans* isolates from Graviera and Galotyri cheeses (Table 5) were similar to the profiles of *E. faecium* isolates from Turkish Beyaz raw ewe milk cheese [53]. Particularly, α-galactosidase activity has been reported as a key differentiating enzymatic reaction between *E. durans* (negative; as strains KE96 = KE100 and KE108 were in Table 5) and its closely-related species *E. hirae* (positive) [54]. Domingos-Lopes et al. [19] concluded that selected LAB strains with β-galactosidase, esterase-lipase, and aminopeptidase activities, with moderate proteolytic activity and high diacetyl production, lacking in β-glucoronidase and β-glucosidase activities, have the potential to be used as adjunct cultures in traditional cheese productions. All Ent+ or m-Ent+ *E. faecium* and *E. durans* Greek cheese isolates of this study fulfill the above technological criteria, except β-galactosidase, in vitro (Table 5). Further studies are needed to evaluate their actual biochemical capabilities as adjunct strains in situ during traditional Greek cheese making trials. So far, only the m-Ent+ *E. faecium* KE82 strain genotype has successfully been applied as a novel antilisterial adjunct culture in experimental Graviera and Galotyri PDO cheeses produced under real factory-scale or pilot-scale manufacturing conditions [55,56].

## 5. Conclusions

In conclusion, apart from the m-Ent+ (A-B-P) *E. faecium* KE82 strain genotype, the m-Ent+ (A-B-P) *E. faecium* KE118, the *ent*A+ *E. faecium* KE64 and GL31, and the *ent*P+ *E. durans* KE100 strain genotypes are also safe for use as cheese adjuncts. All these strain genotypes were shown to be free of risk (virulence gene) determinants and susceptible to vancomycin and most of the other seven antibiotics tested. Notably, the *E. durans* KE100 strain showed stronger aminopeptidase activities (Table 5) and stronger acetoin forming capacity than all tested *E. faecium* strains in vitro [6]. Conversely, the m-Ent+ (*ent*P-*bac*31-*cyl*) *E. durans* KE108 requires further safety validations because it harbors the cytolysin gene [21], probably acquired by horizontal gene transfer from a co-existing *E. faecalis* strain [6,14] in raw milk, the plant environment, or during cooked hard cheese processing [4,5]. Based on the practical experience we have gained since 2015 regarding the routine use of the NisA+ *Lc. lactis* subsp. *cremoris* M78 in commercial Graviera cheese production at the Pappas Bros. traditional dairy (Skarfi E.P.E., Filippiada, Epirus, Greece), it is feasible to circulate autochthonous *Enterococcus* and other wild LAB strains as fresh craft-made adjunct starters by culturing and preserving them in sterile skim milk in domestic freezers (−20 to −30° C) available in small Greek dairies. Factory-scale cheese-making trials are on the way to evaluating the overall biotechnological performance of the present Ent+ or m-Ent+ strains of the *E. faecium/durans* group as adjunct cultures.

## Figures and Tables

**Table 1 microorganisms-09-00777-t001:** Strains of lactic acid bacteria (LAB) used in this study.

LAB Strain/Isolate Code	Origin/Isolation Source	Strain Genotype RAPD-Based	Strain Biotype	Structural Bacteriocin (Enterocin) Gene/s Possessed	Mode of Bacteriocin Activity ^1^	References	GenBank Accession Number
**Autochthonous *Enterococcus* strains under investigation**							
*Enterococcus faecium* GL31	Artisan Galotyri cheese PDO	EF1	1C	Ent+ (*ent*A)	Bacteriostatic	[6,7]	MW709884
*Enterococcus faecium* KE64	Graviera cheese (CSC-free) ^2^	EF2	1B	Ent+ (*ent*A)	Bactericidal	[4,6]	MW644963
*Enterococcus faecium* KE67	Graviera cheese (CSC-free)	EF2	1B	Ent+ (*ent*A)	Bactericidal	[4,6]	ND
*Enterococcus faecium* KE77	Graviera cheese (CSC-free)	EF3	1G	m-Ent+ (*ent*A-B-P-31)	Bactericidal	[4,6]	MW644967
*Enterococcus faecium* KE82	Graviera cheese (CSC-free)	EF4	1D	m-Ent+ (*ent*A-B-P*)*	Bactericidal	[4,6]	MW644969
*Enterococcus faecium* KE118	Graviera cheese (CSC-free)	EF4	1A	m-Ent+ (*ent*A-B-P*)*	Bacteriostatic	[4,6]	ND
*Enterococcus durans* KE96	Graviera cheese (CSC-free)	ED1	2A	Ent+ (*ent*P)	Bactericidal	[4,6]	ND
*Enterococcus durans* KE100	Graviera cheese (CSC-free)	ED1	2A	Ent+ (*ent*P)	Bactericidal	[4,6]	MW644971
*Enterococcus durans* KE108	Graviera cheese (CSC-free)	ED2	2B	Ent+ (*ent*P-*bac*31-*cyl*)	Bacteriostatic	[4,6]	MW644972
**Reference/Control LAB strains**							
*Enterococcus faecium* KE85	Graviera cheese (CSC-free)	Not tested	1D	None (Ent-negative)	None	[4,6]	ND
*Enterococcus faecium* 315VR	Clinical/Human/*van*A*+*	Unknown	NA	Not reported	Not reported	See text	Not reported
*Enterococcus faecalis* ATCC^®^ 29212^TM^	Virulent (*gel*E; *ace*)	Unknown	NA	Not reported	Not reported	[22,23]	ATCC strain
*Streptococcus thermophilus* ST1	CSC strain of natural origin	Not tested	NA	None	None	[24]	Not reported
*Lactococcus lactis* ssp. *cremoris* M78	Greek raw ewe’s milk	M78 = M104	NA	Nisin A (*nis*A)	Bactericidal	[5,27]	JX402634
*Lactiplantibacillus plantarum* H25	Graviera cheese (with CSC)	*Lb. plantarum*	NA	None	None	[4,24]	ND

^1^ The mode of bacteriocin (enterocin) activity refers to the in vitro effects of each strain against *Listeria monocytogenes* no.10 in MRS or M17 broth cocultures at 30 °C for 24 h. ^2^ Commercial starter culture (CSC); naturally fermented and ripened Graviera cheese manufactured with natural undefined yogurt-like starter cultures (CSC-free). Not applicable (NA); not deposited in GenBank yet (ND).

**Table 2 microorganisms-09-00777-t002:** List of primers used in the present study.

Target Gene	Primer Pair	Primer Sequence (5′-3′)	Amplicon Size (bp)	Temp (°C)	References
**Antibiotic resistance structural genes**					
*van*A (Vancomycin)	vanA1	GCTGCGATATTCAAAGCTCA	545	50	Petrich et al., [31]
	vanA2	CAGTACAATGCGGCCGTTA			
*van*B (Vancomycin)	vanB1	ATGGGAAGCCGATAGTCTC	368	50	Petrich et al., [31]
	vanB3	GTTACGCCAAAGGACGAAC			
**Virulence determinants**					
*agg* (Aggregation protein)	Agg-F	AAGAAAAAGAAGTAGACCAAC	1553	53	Espeche et al., [32]
	Agg-R	AAACGGCAAGACAAGTAAATA			
*ace* (Accessory colonization factor)	Ace-F	CAGGCCAACATCAAGCAACA	125	65	Al-Talib et al., [33]
	Ace-R	GCTTGCCTCGCCTTCTACAA			
*esp*A (Enterococcal surface protein)	EspA-F	TTTGGGGCAACTGGAATAGT	407	60	Al-Talib et al., [33]
	EspA-R	CCCAGCAAATAGTCCATCAT			
IS*16* (Transposable element)	IS16-F	CATGTTCCACGAACCAGAG	547	55	Werner et al., [34] EFSA [30]
	IS16-R	TCAAAAAGTGGGCTTGGC			
*hyl* (Hyaluronidase)	HyI-F	ACAGAAGAGCTGCAGGAAATG	276	58	Vankerckhoven et al., [35]
	HyI-R	GACTGACGTCCAAGTTTCCAA			
*gel*E (Gelatinase)	GelE-F	CGAAGTTGGAAAAGGAGGC	372	50	Al-Talibet al., [33]
	GelE-R	GGTGAAGAAGTTACTCTGA			

**Table 3 microorganisms-09-00777-t003:** Hemolytic activity and antibiotic susceptibility of the Ent+ Greek cheese isolates of the *E. faecium/durans* group.

Strain (Isolate)	Type of	Antibiotic Tested (μg/disc)							
	Hemolytic Activity	AMP 10	CHL 30	CIP 5	ERY 15	GEN 10	PEN 10U	TET 30	VAN 30
**Ent+ or m-Ent+ isolates**									
*E. faecium* KE64	α	S (17.5)	S (23.9)	I (18.0)	R (12.1)	S (10.4)	R (13.2)	S (27.5)	S (18.5)
*E. faecium* KE67	α	S (18.4)	S (24.6)	I (16.5)	R (11.7)	S (9.6)	R (12.5)	S (26.9)	S (18.6)
*E. faecium* KE77	α	S (26.4)	S (25.5)	I (18.2)	I (18.5)	S (14.4)	S (22.8)	S (28.1)	S (19.8)
*E. faecium* KE82	α	S (17.3)	S (24.9)	R (12.0)	R (11.0)	S (11.3)	R (11.3)	S (28.3)	S (19.4)
*E. faecium* KE118	α	S (17.0)	S (22.7)	R (13.0)	R (12.7)	S (11.6)	S (14.5)	S (25.9)	S (18.6)
*E. faecium* GL31	α	S (23.6)	S (23.2)	R (14.4)	I (13.5)	S (11.8)	S (19.4)	S (25.3)	S (18.1)
*E. durans* KE96	α/γ	S (22.2)	S (22.1)	I (18.5)	I (21.6)	S (10.6)	S (19.2)	S (22.9)	S (17.8)
*E. durans* KE100	α/γ	S (20.0)	S (21.7)	I (20.4)	I (20.9)	S (10.7)	S (17.1)	S (23.0)	S (18.0)
*E. durans* KE108	α	S (23.4)	S (26.9)	I (19.8)	S (24.0)	S (10.2)	S (19.3)	S (24.0)	S (19.0)
*E. faecium* KE85 (Ent-/control)	α	S (16.6)	S (26.3)	R (12.8)	I (17.4)	S (10.3)	R (13.2)	R (9.6)	S (21.4)
**Virulent *Enterococcus* spp.**									
*E. faecium* 315VR (*van*A+)	β	R (0.0)	S (25.6)	R (0.0)	R (0.0)	R (0.0)	R (0.0)	R (9.5)	R (9.5)
*E. faecalis* ATCC 29212	β	S (23.0)	S (23.7)	I (18.3)	I (15.3)	S (7.6)	S (20.2)	I (16.6)	I (15.6)
**Starter LAB strains**									
*S. thermophilus* ST1	γ	S (28.4)	S (27.2)	S (20.0)	S (27.3)	NT	S (17.3)	S (25.1)	S (19.9)
*Lc. lactis* ssp. *cremoris* M78	γ	S (23.2)	S (24.0)	R (13.2)	S (23.8)	R (0.0)	S (14.2)	R (9.9)	S (17.2)
*Lb. plantarum* H25	α	S (26.0)	S (26.5)	R (9.5)	NT	R (0.0)	S (19.1)	I (17.3)	R (0.0)

Ampicillin (AMP); chloramphenicol (CHL); ciprofloxacin (CIP); erythromycin (ERY); gentamicin (GEN); penicillin (PEN); tetracycline (TET); vancomycin (VAN). The numbers underneath the antibiotics indicate μg per disk or units per disk for penicillin.S, strain susceptible to the antibiotic tested; R, strain resistant to the antibiotic tested; I, the tested strain was of intermediate antibiotic susceptibility according to the Clinical and Laboratory Standards Institute (CLSI) breakpoints [26]; the numbers in parentheses indicate the size (in mm) of the inhibition zone. NT, not tested.

**Table 4 microorganisms-09-00777-t004:** Detection and HPLC-based quantification (mg/L) of histamine (HIS) and tyramine (TYR) formed in vitro by pure cultures of the *Enterococcus* cheese isolates in improved biogenic amine (BA) broth [36] after culturing at 37 °C for 72 h ^1^.

Isolate/Strain	BA Broth without Amino ACIDS Added	BA Broth with 1% Histidine Added	BA Broth with 1% Tyrosine Added
**Ent+ or m-Ent+ isolates**			
*E. faecium* KE64	-/NT	-/NT	++/NT
*E. faecium* KE67	-/NT	-/NT	++/NT
*E. faecium* KE77	(+)/18.4; 35.3	-/74.0	++/1683.8
*E. faecium* KE82	-/NT	-/traces	++/2706.4
*E. faecium* KE118	(+)/NT	-/NT	++/NT
*E. faecium* GL31	(+)/NT	(+)/16.0	++/1247.3
*E. durans* KE96	(+)/NT	(+)/15.2	++/1849.2
*E. durans* KE100	(+)/NT	(+)/15.3	++/1631.6
*E. durans* KE108	-/13.3; 7.3	-/17.8	++/2154.5
*E. faecium* KE85 (Ent-/control)	-	-/15.2	++/2249.1
**Virulent *Enterococcus* spp.**			
*E. faecium* 315VR (*van*A+)	(+)/NT	-/NT	+/NT
*E. faecalis* ATCC 29212	-/NT	-/NT	+/NT
**Starter LAB strains**			
*S. thermophilus* ST1	NG	NG	NG
*Lc. lactis* ssp. *cremoris* M78	-/13.6; 31.6	-/8.1	-/53.8
*Lb. plantarum* H25	-/18.1; 187.1	-/17.0	-/35.1

^1^ Detection of HIS or TYR in BA broth by the indicator’s color change is indicated with the symbols left to each slash; the values given right to each slash indicate the HIS or TYR concentration (mg/L) in the BA sample of each strain in the control BA samples (2nd column), the first concentration value refers to HIS and the second value to TYR; NT = the sample was not tested by HPLC.++ =strong positive reaction (bright dark blue to purple); + =positive reaction (blue to light blue/grey color); (+) = weak positive reaction manifested as browning-greenish of the BA medium;- =negative reaction (yellow to light green color); NG = no growth of the tested strain even in the control medium without added aminoacids.

**Table 5 microorganisms-09-00777-t005:** Enzymatic activity reactions determined by the API ZYM method of the single or multiple enterocin-producing *Enterococcus* Greek cheese isolates ^1^.

	Control Starter or Adjunct LAB Strains ^2^				Ent+ *E. faecium* Isolates					Ent+ *E. durans* Isolates		
Enzyme Assayed for	*S. thermophilus* ST1	*Lc. lactis* M78	*Lb. plantarum* H25	*E. faecium* KE85 Ent-	KE64 KE67	KE77	KE82	KE118	GL31	KE96	KE100	KE108
Alkaline phosphatase	-	5	5	3	2	4	3	3	3	3	3	3
Esterase (C 4)	2–3	3	3	4	3	3	4	3	3	3	4	3
Esterase Lipase (C 8)	-	4	3	3	3	3	4	3	4	4	4	4
Lipase (C 14)	-	-	2	-	-	-	-	-	-	-	-	-
Leucine arylamidase	5	5	5	3	3	4	5	3	4	5	5	4
Valine arylamidase	2	4	5	-	3	2	3	2	3	4	4	4
Cystine arylamidase	2	4	5	-	3	3	4	4	4	4	4	4
Trypsin	-	-	-	-	-	-	-	-	-	-	-	-
α-chymotrypsin	-	3	-	2	1–2	2	4	-	1	-	-	-
Acid phosphatase	3–4	5	5	-	4	4	5	5	5	5	5	5
Naphthol-AS-BI-phosphohydrolase	2	5	4	-	3	3	3	4	3	3	3	3
α-galactosidase	-	-	5	-	-	2	-	-	-	-	-	-
β-galactosidase	4	-	5	-	1–2	-	-	-	-	-	-	-
β-glucuronidase	-	-	2–3	-	-	-	-	-	-	-	-	-
α-glucosidase	-	5	5	-	-	2	-	-	-	-	-	-
β-glucosidase	-	-	4	-	-	-	-	-	-	-	-	-
N-acetyl-β-glucosaminidase	-	-	5	-	-	-	-	-	-	-	-	-
α-mannosidase	-	-	-	-	-	-	-	-	-	-	-	-
α-fucosidase	-	-	-	-	-	-	-	-	-	-	-	-

^1^ Numbers indicate a positive reaction based on the change in color intensity in each API ZYM cupule. According to the manufacturer, the color change was graded using a scale from 1 (extremely weak reaction) to 5 (very strong positive reaction); - indicates a negative reaction, similar to the control. Reactions with a grade of 3 to 5 are considered as clearly positive. ^2^ The four control starter or adjunct LAB strains, ST1, M78, H25, and KE85, were tested comparatively with the Ent+ or m-Ent+ *E. faecium* and *E. durans* isolates in view of their applications as mixed culture preparations in traditional Greek commercial cheese productions.

## Data Availability

The data presented in this study, as well as the bacterial strains used, are available on request from the corresponding author. The 16S rRNA gene sequences of six indigenous Greek cheese isolates are deposited in GenBank; refer to Accession numbers in Table 1.

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
