# Peer review of "Safety Evaluation, Biogenic Amine Formation, and Enzymatic Activity Profiles of Autochthonous Enterocin-Producing Greek Cheese Isolates of the Enterococcus faecium/durans Group"

_microorganisms, 2021, doi:10.3390/microorganisms9040777_

Round 1
Reviewer 1 Report
Dear Authors, the paper is fine and scientifically sound, but very long and not friendly. The paper is plenty of typos and tables are difficult to understand. Thus I suggest to resubmit it after major revision.
Statistic is lacking and PCR results should be shown.
English sometimes seems not fluent and scholastic terms are redundant and turns to mix the fix up.
The title is too long and scholastic. Please revise and find a more friendly way.
The abstract needs to be requalified totally, because the one present has no structure and a minimum of format like “introduction-aims-results-future prospective” should be used.
The conclusion section is not consistent with the results obtained; it is discussed too generally with few links to the findings of the study.
I suggest even to add at least a t-test for significance, and avoid comparing results on values were no stat was made (table 3, 6 , and 7 needs t-test) strongly suggest revising it. Thus, I suggest revising the abstract totally and revising some sentences and terms as follows:
Line 45: antagonistic…please change term
Line 52-53: natural-undefined or commercial-defined
Line 131: written like: “MRS broth (Neogen Culture Media; formerly Lab M, Heywood…” could sound like them are old, and you can affect the interpretation of reproducibility. I think that is important to give to the reader other than microbiologists, a CAS number of the microbiology broth employed, as well as that of supplements.
Line 137: correct typo on °C
Line 137: (Lab M)…please define
Lines 142-145: One of the oldest stock…(see Results). This sentence is not clear, please revise
153-154: enumer-ated comparatively…please remove spelling hyphen and redefine…what does comparatively means in this context? The meaning is fine, but you should explain better…di you apply statistic at least?
154-156: Why did you put the bugs at -20 for 90 days? It is not clear, please specify and give a ref.
163: simple…please remove
164: Listeria monocytogenes no.10…please gives a type strain reference
175: 12 years of frozen…not clear, as it was not clear at lines 142-145
188: The pH…please provide specifics of calibration
198: 2.3.1. Hemolytic Activity…this section needs a reference
200: Bioprepare, Greece…please provide more supplier’s details
210: BioMaxima SA (Poland)… please provide more supplier’s details
227-228: (Biochemistry Laboratory collection)… please provide more supplier’s details
228: Luria agar..do you mean LB? please specify and provide more supplier’s details
230: 2.3.3. Detection of Vancomycin Resistance and Virulence Genes by PCR…This section has no relative results. Please provide them.
234-235: G-STORM GS1, Labtech International Ltd…. please provide more supplier’s details
238: (Germany)… please provide more supplier’s details
256: BA broth medium…please provide more details
287: acetonitrile… please provide supplier’s details
301-303: Thus, all ten isolates….ultra-freezers…please remove this sentence or support it with a ref. Like it is the comparison you talk about is not robust
304: synthetic media…synthetic is redundant, please remove it
305: >9…please change it in “> 9”
305: log…please change in Log10
305: (>9 log CFU/mL)….please report this value as mean ± sd, or remove it
308: pH<5.2…. please change it in “pH < 5.2”
315: pH<5.5…. please change it in “pH < 5.5”
Table 3: D-log…D or delta?
Table 3: cfu…change in CFU
Table 3: -20°C…please correct
Table 3: ml…change in mL
Table 3: log… please change in Log10
Table 3: 37 °C;20 h…please correct
Table 3: NT…please define
Table 3: (+)/-…???
Table 3: PLEASE correct all typos, the table and the footnotes are plenty
325: (<0.7 log reduction)…please provide the right unit format and typing and give a value as mean ± sd
325-329: no stat is applied and thus no comparison can be made
333:-336: Most importantly….shown)..please remove this sentence or report data.
353: pH<6.0
Table 6: check for typos
458-465: This time comparison on a statistical level can be easily made with a t-test at least. Please now that you have numbers, statistic is fundamental for any comparison.
466-468: None…shown)…this feature is too important not to have data shown…please provide a supplementary material or write in the text some evidences or delete the sentence.
Table 7: has many typos and editing errors. For example,it is not clear were to read the results of hemolytic activity (Check for typos too). The Greek letters of the hemolytic activity must goes in lowercase.
Table 7: NT…please define
484: Also, as it was expected…please avoid this kind of sentence if there’s no ref. It also reduces the impact of your testing.
Table 8: is very confusing, for each case there are two values in a column, but it is not specified how to interpret them.
Table 8: check for typos in title, in the body of the table and in the footnotes. Please uniform the fashion you have used in the different tables
Table 8: n.d. and NT. Please uniform the style, as well as defining better this NT
Table 8: (mg/liter)…change in mg/L
Table 8: Please include significance values
Table 8: has to be redone
504: hospital strains…??? Please be more accurate in definition
506: feed additives in animal nutrition….??? Please give explanation of the EFSA document or reader could think you misunderstood.
506-507: this requirement may be extended to E. faecium strains intended for use in foods…who says so? Feed and food are totally different
508: virulent traits…I am not sure that resistance to antibiotics is a virulent trait.
512-519: The comparison could be fitting, but not the length in explaining others results.
520-531: Please revise the sentences in a clearer way, avoid including all that commas and secondary sentences. Please be shorter in explaining a concept.
552-558: In general…cheeses. Please make shorter sentence splitting it in two. The way of discussing other results is very confusing.
558-560: Especially, industry…I don’t understand the reason to discuss E. italicus, since you have no such isolate in your list.
564: basic research data…???
564-569: Notably…traits [62, 65]. This sentence is pretty chaotic and there’s no link with your findings
571-572: It is noteworthy… (Table 1)…THE verb tense is wrong, please revise
Line 606: 379-4339, 610, and 601-4986 mg/l…avoid giving others’ value if you do not compare those with yours.
617-618: >200 mg/kg…add a space after >
619: >1,000 mg/kg… add a space after >, report kg as Kg, and avoid the comma in the value.
632-633: Further in-depth multi-factorial research studies…???

Author Response
Dear Reviewer 1
Please see attached file with our responses to your comments

Reviewer 2 Report
General comment:
The manuscript by Tsanasidou et al. reports whether the antagonistic strains of the E. faecium/durans group are completely harmless, resistant and stable to freezing, and metabolically suitable to be applied as co-starter or protective adjunct cultures in traditional Greek cheese technologies. The methods are well written and the results are clearly presented and discussed. According to the presented results, novel autochthonous enterocin-producing strains of the Enterococcus faecium/durans group from the Greek cheese were characterised, what is the main scientific contribution of the manuscript. However, a few of improvements are needed in the terms of clarity of the manuscript. Therefore, the manuscript is not acceptable for publication in its present state.
Specific comments:
In recent publications of the same authors, scientific subjects were similar to the aim of this manuscript. Maybe this is the reason why results related to the spectrum of bacteriocinogenic activity of examined autochthonous strains were not shown. Although adjunct cultures can contribute to cheese quality with novel flavor metabolites, one of the most important trait of adjunct cultures is antagonistic activity against spoilage microorganisms to ensure safe dairy products of high microbiological quality. It was only examined “the in vitro effects of each strain against Listeria monocytogenes”, but Listeria monocytogenes is not the only one potential spoilage bacterium in the fermented milk products. Namely, if enterocins produced by examined strains do not inhibit potential milk pathogens, their application as adjunct cultures in Greek cheeses production is questionable, regardless all other results obtained. Usually used starter cultures already have all metabolic activities necessary for cheeses production, but they don’t possess specific antibacterial activity which is connected with bacteriocin production of adjunct starter strains. If the results of antagonistic activities of examined autochthonous strains as potential adjunct starter cultures are already published, this must be mentioned in the section Discussion. Otherwise, additional experimental data are needed to confirm the antagonistic activity against food spoilage microorganisms.
Line 172 Table 1 does not contain “The entire carbohydrate (sugar) fermentation profiles of the Ent+ Enterococcus cheese isolates (Table 1)” as authors stated in the text. The sentence must be corrected.
Line 182 There is again confusion about content of Table 1. In the sentence “All dairy LAB in Table 1 were tested for their milk fermentation…” The sentence must be corrected.
Table 4. as well as Table 5. can be removed as supplementary information because results were obtained by standard API methods.
Table 7 Results of Hemolytic activity are not in the column where the title is mentioned! Used symbols, related with these results, are not explained under the table!
Author Response
Dear Reviewer 2
Please see attached file with our responses to your comments

Reviewer 3 Report
The use of institutional email is a necessity.
Abstract should also contain at the end the conclusions of the study.
Extensive inappropriate citation. 32 ref only in the introduction. Please reduce them. Also please reduce the introduction part.
I cannot find the aim of the present study. Although I may understand it but I cannot find it in the text.
The authors report that the present study is a follow-up study. Follow up of which?? There are many previous studies of the same group using the same cultures.
Extensive self-citation. Reduce.
Lines 142-245.????? Delete.
Usually when separate sections of Results and Discussion occur, no references are added in results. In this section only the results are presented and a critical discussion with references occurs only in Discussion section. In this study this is not the case. Please revise.
Section 2.2.1. Please explain the significance of this experiment. There is only the presentation of the results in the section “Results”, but there is no discussion. This part has non practical meaning. All these experiments should be deleted. In addition, I would like to ask the authors since they had isolate the microorganisms 12 years ago why they made the experiments now? So many studies with these strains they have published but now made these safety experiments?
Please explain short and long term storage. This is very complicated in the text.
In my opinion the work is not well organized. It seems like different experiments without meaning gathered in order to be published. Long term storage (12 years????) it was designed? Why not at 10, 15 etc? Short term freezing in milk??? Different temperatures of freezing -20oC?? -30oC ?? applied. Why? What about the traditional deep freezing -70oC???
-30oC why? Is this appropriate?
Short term storage for 90 days. Why? Why not for 60 days or 30 days or 120 days?
There is extensive discussion but the majority is the presentation of other works. The authors should reduce this part and keep only relevant information.
In general please rewrite the manuscript. Present the novelty, originality aim in the text. Avoid extensive citation and self-citation.
Author Response
Dear Reviewer 3
Please see attached file with our responses to your comments

Round 2
Reviewer 1 Report
Dear Authors, thanks for the revision, even if I didn't understand the logic benath that feed and food are complementary....anyhow I am satisfy with your revision and the paper for me can be accepted
ther's still to consider editing of table 2 and revising a sentence that maybe is lacking a verb
line 384:
Notably, the cytolysin gene presented in our m-Ent+
E. durans KE108 strain genotype also (Table 1).
Sincerely
LN
Author Response
Dear Reviewer 1
Attacjhed please find a PDF file with our respones. Thank you.
John Samelis

Reviewer 2 Report
The authors accepted all the requested remarks and answered all the issues, what contributed to the overall manuscript improvement. Therefore, the manuscript is appropriate for the publication.
Author Response
Dear Reviewer 2
Attached please find a PDF file with our responses.
Thank you.
John Samelis
